# Melanoma Mortality Trends in 28 European Countries: A Retrospective Analysis for the Years 1960–2020

**DOI:** 10.3390/cancers15051514

**Published:** 2023-02-28

**Authors:** Paweł Koczkodaj, Urszula Sulkowska, Joanna Didkowska, Piotr Rutkowski, Marta Mańczuk

**Affiliations:** 1Cancer Epidemiology and Primary Prevention Department, Maria Sklodowska-Curie National Research Institute of Oncology, 02-781 Warsaw, Poland; 2National Cancer Registry, Maria Sklodowska-Curie National Research Institute of Oncology, 02-781 Warsaw, Poland; 3Department of Soft Tissue/Bone Sarcoma and Melanoma, Maria Sklodowska-Curie National Research Institute of Oncology, 02-781 Warsaw, Poland

**Keywords:** melanoma, mortality, cancer prevention, cancer epidemiology, Europe, EU

## Abstract

**Simple Summary:**

Efficient health interventions in individual populations should be based on evidence-based data. Our study provides new insights on melanoma (except primary uveal melanoma) mortality trends in European Union (EU) countries, as well as in three non-EU countries, from a wide-time perspective (1960–2020), by sex and age groups. To the best knowledge, our study is the most up-to-date and broad analysis of melanoma mortality in Europe. It can be used as a reference for public-health actions aiming at a national and international melanoma-mortality reduction. Investigated melanoma-mortality trends vary in individual countries and age groups; however, a highly concerning phenomenon—an increasing melanoma mortality in both sexes—was observed in the case of 7 countries for the younger age group and 26 countries for the older one. There is a need for coordinated systemic actions to address this issue.

**Abstract:**

Background: In 2020, in 27 European Union (EU) Member States, melanoma accounted for 4% of all new cancer cases and 1.3% of all cancer deaths, making melanoma the fifth most common malignancy and placing it in the 15 most frequent causes of cancer deaths in the EU-27. The main aim of our study was to investigate melanoma mortality trends in 25 EU Member States and three non-EU countries (Norway, Russia, and Switzerland) in a broad time perspective (1960–2020) in a younger (45–74 years old) vs. older age group (75+). Methods: We identified melanoma deaths defined by ICD-10 codes C-43 for individuals aged 45–74 and 75+ years old between 1960–2020 in 25 EU Member States (excluding Iceland, Luxembourg, and Malta) and in 3 non-EU countries—Norway, Russia, and Switzerland. Age-standardized melanoma mortality rates (ASR) were computed using the direct age-standardization for Segi’s World Standard Population. To determine melanoma-mortality trends with 95% confidence intervals (CI), Joinpoint regression was applied. Our analysis used the Join-point Regression Program, version 4.3.1.0 (National Cancer Institute, Bethesda, MD, USA). Results: Regardless of the considered age groups, in all investigated countries, in general, melanoma standardized mortality rates were higher for men than women. Considering the age group 45–74, the highest number of countries was characterized by decreasing melanoma-mortality trends in both sexes—14 countries. Contrarily, the highest representation of countries in the age group 75+ was connected with increasing melanoma-mortality trends in both sexes—26 countries. Moreover, considering the older age group—75+—there was no country with a decreasing melanoma mortality in both sexes. Conclusions: Investigated melanoma-mortality trends vary in individual countries and age groups; however, a highly concerning phenomenon—increasing melanoma-mortality rates in both sexes—was observed in 7 countries for the younger age group and in as many as 26 countries for the older age group. There is a need for coordinated public-health actions to address this issue.

## 1. Introduction

The incidence of melanoma is driven mainly by overexposure to a well-known single risk factor, ultraviolet radiation (UV), emitted naturally by the sun and artificially by sunbeds and other tanning devices. Even though knowledge of UV-preventive measures seems to be expected, in the past decades, the global melanoma incidence has been increasing [1]. According to the Globocan data, in 2020, the number of new melanoma cases was estimated at 324,635 in both sexes (age-standardized rate—ASR: 3.4/100,000). The same year, melanoma contributed to 57,043 deaths worldwide (ASR: 0.56/100,000) [2]. Additionally, in 2020, in 27 European Union (EU) Member States, melanoma accounted for 4% of all new cancer cases and 1.3% of all cancer deaths, making melanoma the fifth most common malignancy and placing it in the group of the 15 most frequent causes of cancer deaths in the EU-27 [3].

Our study aimed to investigate melanoma-mortality trends in 25 EU Member States and three non-EU countries (Norway, Russia, and Switzerland) in a broad time perspective (1960–2020) among a younger vs. older age group. We also attempted to identify the patterns and regularities in the melanoma-mortality trends and presented melanoma-mortality rates in the investigated countries.

To the best of our knowledge, our study is the most up-to-date and broad analysis of melanoma mortality in Europe.

## 2. Materials and Methods

The analyzed mortality and population data were obtained from the World Health Organization (WHO) Mortality Database [4] and Eurostat [5]. We identified melanoma deaths defined by ICD-10 codes C-43 for individuals aged 45–74 and 75+ years old between 1960–2020 in 25 EU-Member States (excluding Iceland, Luxembourg, and Malta) and three non-EU countries—Norway, Russia, and Switzerland. The selection of the age groups was based on a preliminary analysis of mortality datasets (meager melanoma-mortality rates in age groups younger than 44 years old) as well as taking into consideration biological features of melanoma—an average latency period of about 20–40 years (time from exposure to melanoma occurrence) [6]. Due to the small populations in Iceland, Luxembourg, and Malta, we decided not to present melanoma-mortality trends for those countries. This data could be misleading and potentially contribute to wrong conclusions. Moreover, in selected countries, we have included the United Kingdom, as Brexit formally took place in 2020 and had no relevant impact on analyzed data in this particular year.

Age-standardized melanoma-mortality rates (ASR) were computed using the direct age-standardization for Segi’s World Standard Population [7]. We used the Joinpoint Regression Program (version 4.3.1.0, National Cancer Institute, Bethesda, MD, USA) to analyze melanoma-mortality time trends. Joinpoint regression was applied to determine melanoma-mortality trends with 95% confidence intervals (CI) [8]. The best-fitting model was selected with permutation tests, with an overall significance level of 0.05. Our study was performed according to the Strengthening the Reporting of Observational Studies in Epidemiology (STROBE) guidelines [9].

### Limitations of the Study

Taking into consideration the long period of the observation (60 years), some of the data (mainly from the earliest years) may not fully reflect the total size of the investigated phenomenon, as the process of data collection and its standardization has been improved over the years and is, in the case of some countries, still ongoing.

## 3. Results

Regardless of the considered age groups, in all investigated countries, melanoma standardized mortality rates were in general higher for men than women.

Considering the age group 45–74, the highest number of countries was characterized by decreasing melanoma-mortality trends in both sexes—14 countries. Contrarily, the highest representation of countries in the age group 75+ was connected with increasing melanoma-mortality trends in both sexes—26 countries. Moreover, considering the older age group—75+—there was no country with a decreasing melanoma mortality in both sexes (Table 1).

### 3.1. Age Group: 45–74 Years Old

In the considered age group, seven investigated countries were characterized by an increasing melanoma mortality, with the highest age-standardized mortality rates in Slovenia for both men, and women. In the case of Greece and Lithuania, we observed a constant and stable increase in melanoma-mortality trends, which was, however, more dynamic in women in Greece and men in Lithuania. When discussing other countries, we investigated breakthrough years where the pace of increase in melanoma mortality slowed down—in 1982 in Italy (in both sexes) and Ireland (in women); in 1986 in Slovenia (in both sexes); in 1994 and 1997, respectively, for men and women in Portugal and in 2001 in Slovakia (in men) (Figure 1).

In the analyzed age group, the highest proportion of countries (15) showed decreasing melanoma-mortality trends in both sexes. Among those countries, the breakthrough years, when melanoma-mortality trends started to decline, varied from the early 1980s through the early and late 1990s to the second decade of the 21st century, when the highest proportion of countries was noted to have a melanoma-mortality decrease. Moreover, in the vast majority of the analyzed countries, a melanoma-mortality decrease occurred earlier in the case of women compared to men, except for the Czech Republic, Germany, Norway, Russia, and Spain (Figure 2).

In the analyzed age group, we also investigated two countries (Bulgaria and France) with a melanoma-mortality increase in women and a decrease. In the case of women, we observed a stable increase in Bulgaria. In France, the pace of growth slowed down in 1984 (Figure 3). In both countries, a decline started around 2012. 

The last group of countries underwent a decrease in melanoma mortality in women and an increase in men. In the case of Croatia and Finland, we observed the breakthrough years in men, respectively, to be in 2001 and 1970, when the pace of melanoma mortality decreased. In other countries, the increase in men was stable. We also observed that the decrease in melanoma mortality in Finish women started much earlier—in the early 1970s—compared to other countries from this group (Figure 4).

In Figure 5, we presented the latest available values of the age-standardized melanoma-mortality rates in women and men aged 45–74. In women populations, melanoma mortality rates varied from 1.6 in Greece to 4.7 in Slovenia. The highest proportion of rates was between 3.0–3.9 (12 countries) and 2.3–2.9 (10 countries). The most increased mortality was characterized for Scandinavian women (except for Slovenia, for which the highest rate was at 4.7)—Netherlands: 4.2; Denmark: 4.2; Norway: 4.0; Sweden: 3.9. The three lowest values were linked to Greek, Spanish, and Romanian women, respectively, at 1.6, 1.8, and 2.3.

In general, the values of the current age-standardized melanoma-mortality rates are higher in men than women. The values of melanoma rates varied from 3.0 in Spain to 8.6 in Croatia. The highest proportions of countries were characterized by a subsequent range of melanoma-mortality rate values: 3.0–3.8 (6 countries); 4.0–4.8 (7 countries); 5.1–5.6 (7 countries). Contrary to women, only one Scandinavian country—Norway (7.7)—was among those with the highest melanoma mortality. The three lowest rates were linked to Spain, Greece, and Portugal, with rate values of, respectively, 3.0, 3.5, and 3.6.

### 3.2. Age Group: 75+ Years Old

As much as 26 of 28 investigated countries noted increasing melanoma-mortality trends in men and women in the age group of 75 years old and older. Contrary to the youngest age group (45–74), the pace of the melanoma-mortality increase—particularly among men—was much more dynamic, reaching exceptionally high values of melanoma-mortality standardized rates in a relatively short time in Estonia, Ireland, Norway, and Slovenia (Figure 6). On the other hand, some of the investigated countries underwent a much lower and less dynamic increase in the same period: for example, Belgium, Bulgaria, Greece, and Romania. Almost in every investigated country, we noted a constant rise in melanoma mortality among men, except Austria, France, and Switzerland (two times), with visible breakthrough years (respectively, in 1988, 1968, 1975, and 1992), when the pace of increase slowed down. Similarly, in the case of women, we observed this phenomenon in only a few countries—Austria (1993), Belgium (1970), Czech Republic (1995), Finland (1967), Italy (1980), Slovenia (1990), and Switzerland (1986) (Figure 7). In two countries, Hungary and Portugal, we investigated the melanoma-mortality decrease in women and increase in men (Figure 8). In Hungary, trends in men and women followed almost the same pattern until the early 1980s, when melanoma mortality started to increase among men and decrease in women. In Portugal, trends in both sexes were similar until the early 2000s. After that time, the pace of the melanoma-mortality increase in women slowed and finally decreased in 2013. Unlike the women population, we observed a constant increase in melanoma mortality among Portuguese men.

In Figure 8, we present the latest available age-standardized melanoma-mortality rates for women and men. Compared with the younger age group—45–74 years old—those investigated are much higher, regardless of gender. In women, rates varied from 6.8 in Spain to 27.1 in Slovenia. Apart from Spain, the lowest melanoma mortality was linked to women in Portugal: 7.1; Greece: 7.4; Bulgaria: 7.9; and Romania: 7.9. Besides Slovenia, the following three highest values occurred in Scandinavian countries: Norway: 22.8; Sweden: 20.4; and Denmark: 20.1. Again, similarly to the younger age group, mortality is higher among men than women. However, differences between genders in the given countries are higher in the age group 75+. In men, melanoma-mortality rates varied from 14.1 in Portugal and Russia to 53.4 in Norway. The second highest value was investigated in Slovenia and the third in Denmark. Besides Portugal and Russia, the lowest mortality rates were linked to Belgium: 14.4; Romania: 14.8; and Greece: 14.9.

## 4. Discussion

The results of our study confirmed a general regularity in cancer-mortality increases connected with aging. So far, many studies have described increasing age as one of cancer patients’ most substantial prognostic factors [10]. Additionally, in our research, higher melanoma-mortality rates were linked to the older age group—75 years old and older—in both men and women. In their study, Ribero et al. [11] indicated that in the case of melanoma as well, aging is an important prognostic factor—increasing age is connected with the worst survival prognosis, which may be related to a lower penetration of awareness campaigns, leading to diagnosis in more advanced stages as well as a lower use of modern systemic therapies due to comorbidities commonly existing in this age group.

Further, our study results showed that the highest melanoma mortality is more characteristic for Nordic countries, such as Norway, Sweden, Netherlands, and Denmark, irrespective of the age group and gender. Similar to our results, Robsahm et al. [12] investigated an exceptionally high melanoma mortality in Norway, particularly in men aged 70 and older. The authors suggested that the possible reason for this phenomenon was a late diagnosis. Other study results concerning melanoma mortality in Sweden also align with our outcomes, showing high melanoma-mortality rates. Additionally, the authors suggested a late diagnosis as the main reason in this case. Still, they also indicated gender (higher burden in men) and a higher age at diagnosis as other essential factors [13]. Moreover, a population-based Netherlands Cancer Registry data analysis showed that melanoma-survival prognosis rates are more favorable for women [14]. Our study also proved a higher melanoma mortality among men compared to women, which is in line with the outcomes of another study. The background of this phenomenon still needs to be clarified. However, there is some evidence that the most probable explanation is genetic and hormonal differences between genders [15].

On the other hand, Nordic countries have the lowest annual sunlight hours; for example: Denmark—1618 h/year; Norway—1645 h/year; and Sweden—1981 h/year. Meanwhile, these numbers in Spain, Greece, and Portugal are, respectively, at 2435 h/year, 2804 h/year, and 3044 h/year. Paradoxically, sunnier countries generally demonstrated a lower melanoma mortality in both investigated age groups. In their study, Shipman et al. [16] presented similar results. The higher burden of melanoma in northern countries could be explained by the frequent use of sunbeds, confirmed by the *Euromelanoma* campaign data analysis [17]. Moreover, the skin of the people who originally lived in northern latitudes has a higher sun sensitivity, which makes them more prone to artificial and natural ultraviolet-overexposure health consequences [18].

Interestingly, Slovenia has the highest melanoma-mortality rates among all European women in both age groups and the second highest among men aged 75+. Considering *Euromelanoma* data on sunbed-use prevalence in 30 European countries, only 6% of Slovenians use these devices (in comparison, in Belgium—26.5% or Latvia—25.2%). Additionally, Slovenia can be considered a southern country. Therefore, Slovenian inhabitants should demonstrate an individually lower sun sensitivity. Moreover, in 2016, in Slovenia, almost all innovative medicines for metastatic melanoma were available—registered and reimbursed (except for Nivolumab—lack of reimbursement) [19]. Despite these facts, we investigated melanoma mortality in this country in both sexes and age groups. Partially, the explanation of this phenomenon could be connected to the lack of tanning-bed legislation in this country. Slovenia is one of the few EU-Member States without any law banning the usage of sunbeds by minors [20]. This argument is important because sunburns in children significantly increase the risk of melanoma development in adult life [21,22]. Even though the declared use of sunbeds in Slovenia can be considered low, there is no safe UV-radiation level from sunbeds [23].

Moreover, data published by the European Observatory on Health Systems and Policies [24] showed that Slovenia has a low number of physicians (fewer than the EU average) with the highest proportion of nurses. At the same time, according to Kandolf et al., as much as 20% of Slovenian patients with suspicious changes on their skin are examined by physicians [25]. Only in 2019 did some of the tasks start to be shifted from medical doctors to nurses. Meanwhile, shortages of healthcare professionals can be perceived as one of the main barriers to the efficient performance of primary and secondary cancer-prevention actions [26].

Apart from Slovenia, in the younger age group of 45–74 years old, we investigated the melanoma-mortality increase in both sexes in six other countries: Greece, Ireland, Italy, Lithuania, Portugal, and Slovakia (Table 1). Older age is connected with the worst survival prognosis; however, in the younger population’s case, other significant factors may play a vital role in the melanoma-mortality increase. The efficiency of healthcare systems, determined by public-health investments, should be considered as one such factors [27]. In recent years, health expenditures per capita in all the countries mentioned above remained below the rate in the EU. Furthermore, there was no meaningful increase in health expenditures in Greece, Italy, and Slovakia in previous years. In Lithuania, Portugal, and Slovenia, the increase was marginal. Only in the case of Ireland was the spending on health per capita close to the EU average; however, in this country, inpatient care is less accessible than in other EU countries. Additionally, in Ireland, in 2019, spending on prevention was below the EU average [28,29,30,31,32,33].

On the other hand, Italy was second among the countries with the highest value of current healthcare expenditure on preventive care (after the UK). Similarly, Slovenia was among the countries above the EU average, which could translate into lower and decreasing melanoma incidence and mortality rates in these countries [34]. However, Eurostat data did not distinguish if provided funds were specifically spent on cancer prevention (particularly, melanoma preventive actions). Nevertheless, national-health expenditures seem to be an essential factor impacting melanoma survival, which was also investigated by Forsea et al. in their study (strong correlation between melanoma MIR—mortality-to-incidence ratios—and national health expenditure levels) [35].

Finally, access to innovative treatment options is often perceived as the most vital factor impacting cancer-survival rates and melanoma. According to a study by Kandolf et al. [19], in the vast majority of European countries, melanoma patients are treated with the use of innovative medicines (>90% of patients as of 2018). The only exceptions, where percentages are lower than 90%, are countries such as Portugal—30–50% of melanoma patients; the United Kingdom—70–90%; Bulgaria—50–70%; the Czech Republic—70–90%; Lithuania—30–50%; Romania—50–70%; and Russia—<10%. The above results are partially in line with the outcomes of our study. On the one hand, the Czech Republic and Lithuania are among the countries with the highest melanoma-mortality rates and the worst access to innovative treatment options. On the other hand, countries such as Bulgaria and Romania also have the worst access to innovative treatment options. Still, they are characterized by one of the lowest melanoma-mortality rates.

## 5. Conclusions

Investigated melanoma-mortality trends vary in individual countries and age groups; however, a highly concerning phenomenon—increasing melanoma-mortality rates in both sexes—was observed in 7 countries for the younger age group and in as much as 26 countries for the older age group. Undoubtedly, a higher cancer mortality is related to older age. However, this phenomenon raises a question regarding cancer care that is available for older people, which should be a subject of further research.As in most countries, melanoma-mortality-rate values are higher for men, and the pace of increase is much more rapid for men; one should consider putting more emphasis on gender-tailored preventive actions. Likewise, in the case of age groups, melanoma mortality increases with age.The results of our study suggest that there is not only one universal factor affecting melanoma mortality for all investigated European countries. Rather, we should define a mixture of the most significant factors that substantially impact individual European populations with different intensities. We assume that the most substantial factors are (starting with the most important) aging, the efficiency of healthcare systems, good access to innovative treatment options, and UV overexposure (mainly caused by sunbeds).Despite the high availability of innovative treatment options for melanoma cancer patients (in the vast majority of investigated countries, >90% of melanoma patients had access to innovative treatment), not all of these countries have undergone decreasing melanoma trends. Moreover, some of them have the highest current melanoma-mortality-rate values. This phenomenon could be possibly explained by a period that is too short since the given treatment options have been available or/and a worse efficiency of cancer care and the healthcare system in individual countries. Moreover, an assurance of innovative medicines is insufficient for a melanoma-mortality decrease during the staff shortages, underfinancing, or organizational difficulties in healthcare that many investigated countries have faced for decades. However, these hypotheses should be the subject of further studies.Considering the obtained results, primary and secondary actions aiming for the prevention and early detection of melanoma seem to be notably underestimated by policymakers and not effectively performed in many investigated countries—in most cases, primarily the fact that the early symptoms of melanoma can be easily recognized.There is a need to strengthen the efforts of medical societies to promote education on melanoma prevention, as health professionals have reliable knowledge and the highest trust among patients. We can assume that, as in the case of other highly preventable cancers (e.g., cervical cancer, where, similar to melanoma, a single risk factor plays a crucial role in elevating the risk of the disease), mortality reduction demands much broader coordinated public-health actions than primary prevention alone. The effectiveness of these systemic and individual actions will determine the success of such strategic initiatives as Europe’s Beating Cancer Plan.

## Figures and Tables

**Figure 1 cancers-15-01514-f001:**
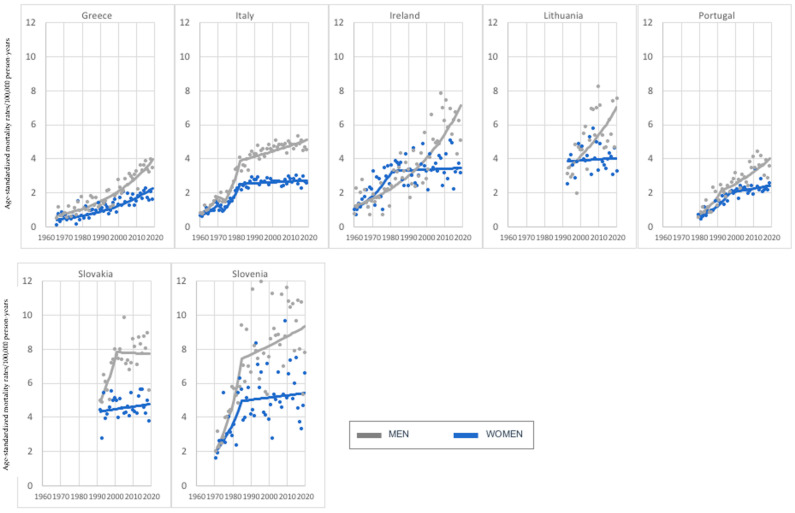
Melanoma mortality trends—7 countries with investigated increase in both sexes, age group 45–74 years old.

**Figure 2 cancers-15-01514-f002:**
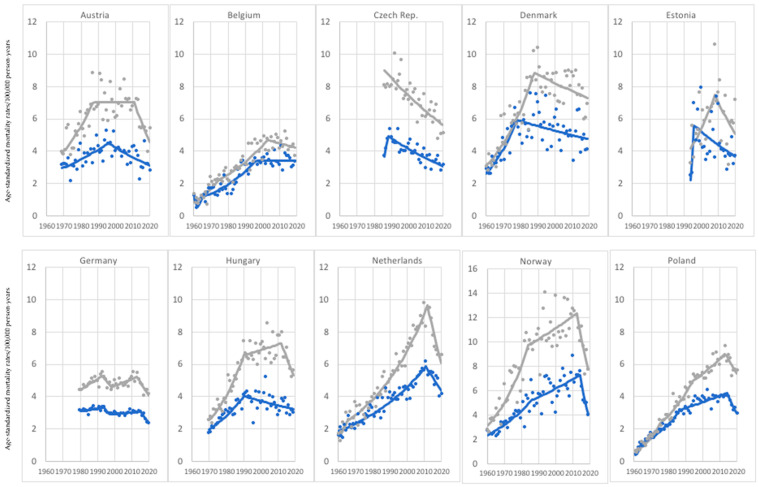
Melanoma mortality trends—15 countries with an investigated decrease in both sexes, age group 45–74 years old.

**Figure 3 cancers-15-01514-f003:**
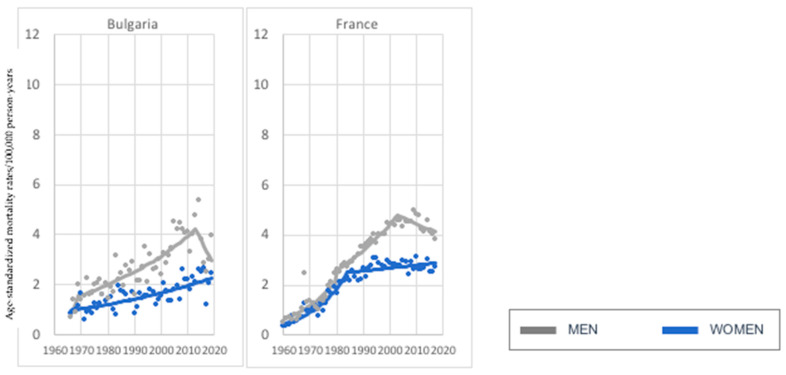
Melanoma mortality trends—2 countries with an investigated increase in women and decrease in men, age group 45–74 years old.

**Figure 4 cancers-15-01514-f004:**
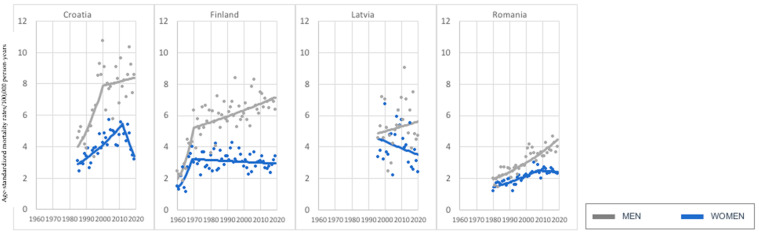
Melanoma mortality trends—4 countries with an investigated decrease in women and increase in men, age group 45–74 years old.

**Figure 5 cancers-15-01514-f005:**
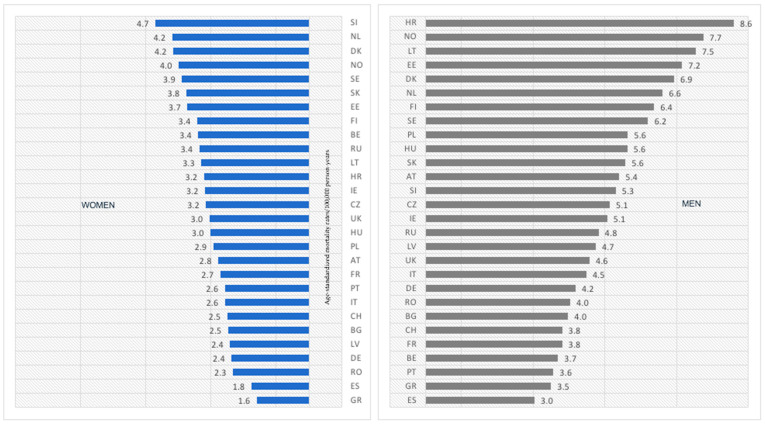
Age-standardized melanoma-mortality rates in women and men aged 45–74 years old in 2020 or the last available year (2018 or 2019) in 28 European countries (*SI—Slovenia; NL—the Netherlands; DK—Denmark; NO—Norway; SE—Sweden; SK—Slovakia; EE—Estonia; FI—Finland; BE—Belgium; RU—Russia; LT—Lithuania; HR—Croatia; IE—Ireland; CZ—Czech Republic; UK—United Kingdom; HU—Hungary; PL—Poland; AT—Austria; FR—France; PT—Portugal; IT—Italy; CH—Switzerland; BG—Bulgaria; LV—Latvia; DE—Germany; RO—Romania; ES—Spain; GR—Greece*).

**Figure 6 cancers-15-01514-f006:**
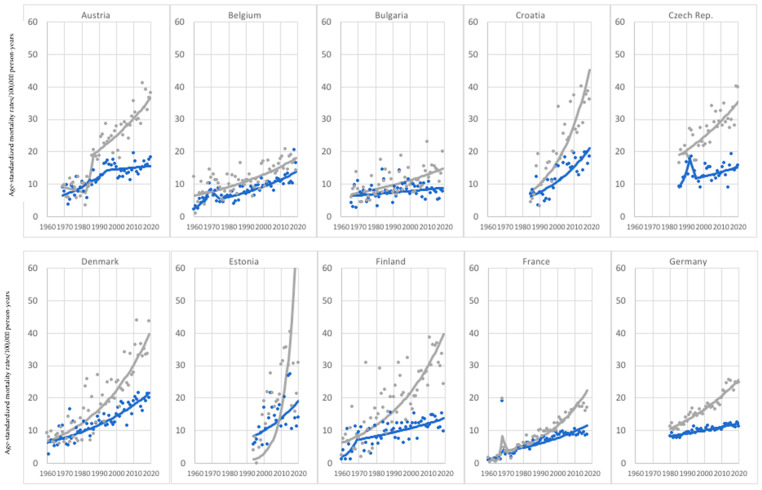
Melanoma mortality trends—26 countries with investigated increase in both sexes, age group 75+ years old.

**Figure 7 cancers-15-01514-f007:**
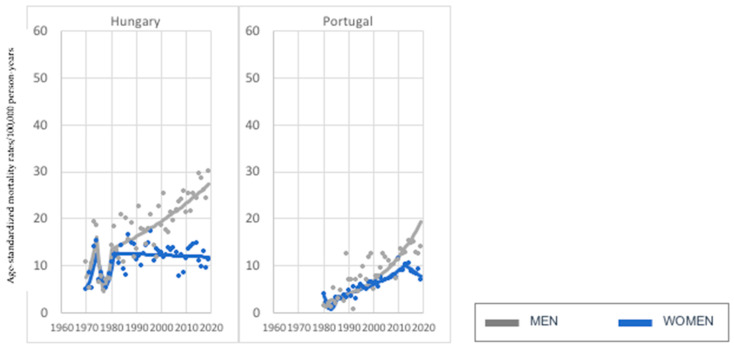
Melanoma mortality trends—2 countries with an investigated decrease in women and increase in men, age group 75+ years old.

**Figure 8 cancers-15-01514-f008:**
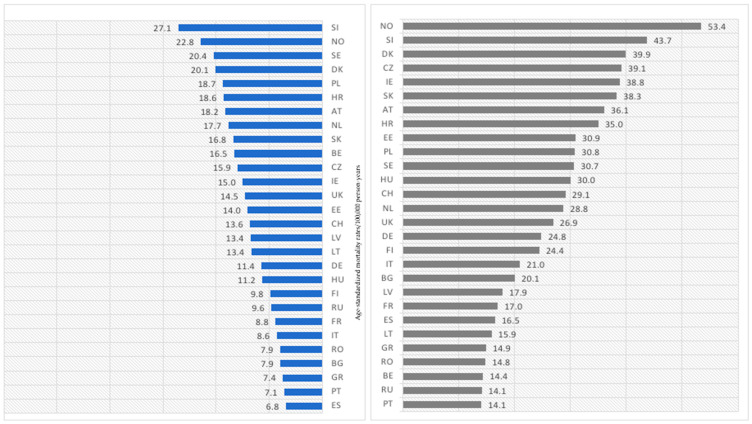
Age-standardized melanoma-mortality rates in women and men aged 75+ years old in 2020 or the last available year (2018 or 2019) in 28 European countries (*SI—Slovenia; NO—Norway; SE—Sweden; DK—Denmark; PL—Poland; HR—Croatia; AT—Austria; NL—the Netherlands; SK—Slovakia; BE—Belgium; CZ—Czech Republic; IE—Ireland; UK—United Kingdom; EE—Estonia; CH—Switzerland; LV—Latvia; LT—Lithuania; DE—Germany; HU—Hungary; FI—Finland; RU—Russia; FR—France; IT—Italy; RO—Romania; BG—Bulgaria; GR—Greece; PT—Portugal; ES—Spain*).

**Table 1 cancers-15-01514-t001:** Melanoma-mortality-trends matrix for European countries by age group, sex, and country.

Current Mortality Trend	Age Groups	
	45–74	75+
Increase—both sexes	Greece, Italy, Ireland, Lithuania, Portugal, Slovakia, Slovenia (7)	Austria, Belgium, Bulgaria, Croatia, Czech Republic, Denmark, Estonia, Finland, France, Germany, Greece, Ireland, Italy, Latvia, Lithuania, Netherlands, Norway, Poland, Romania, Russia, Slovakia, Slovenia, Spain, Sweden, Switzerland, United Kingdom (UK) (26)
Decrease—both sexes	Austria, Belgium, Czech Republic, Denmark, Estonia, Germany, Hungary, Netherlands, Norway, Poland, Russia, Spain, Sweden, Switzerland, United Kingdom (UK) (15)	None (0)
Increase in women, decrease in men.	Bulgaria, France (2)	None (0)
Decrease in women, increase in men.	Croatia, Finland, Latvia, Romania (4)	Hungary, Portugal (2)

## Data Availability

Data are available upon reasonable request from the corresponding author and the publicly available World Health Organization and Eurostat datasets.

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
