# Peer review of "Melanoma Mortality Trends in 28 European Countries: A Retrospective Analysis for the Years 1960–2020"

_cancers, 2023, doi:10.3390/cancers15051514_

Round 1

Reviewer 1 Report (Previous Reviewer 3)

Nothing to add to the reviewed version.

Author Response

Dear Reviewer,

Thank you for your comments. We introduced changes in our manuscript accordingly.

Sincerely,

Authors

Reviewer 2 Report (Previous Reviewer 2)

this is an improved version, looks good. 

Author Response

Dear Reviewer,

Thank you for comments. We introduced changes in our manuscript accordingly.

Sincerely,

Authors

This manuscript is a resubmission of an earlier submission. The following is a list of the peer review reports and author responses from that submission.

Round 1

Reviewer 1 Report

Authors present an analysis of melanoma mortality in the EU based on WHO and Eurostat. They show increasing melanoma mortality over decades of data.

Major Comments:

-The paper needs English editing for flow, as it reads quite clunky.

-The abstract starts with describing melanoma as "one of the most preventable cancers". Where risk can be reduced with sun exposure, stating this in the introduction seems to place the patient at blame and be rather judgemental.

-Key paper in this field is the note of increasing melanoma diagnoses (N Engl J Med 2021; 384:72-79), but it is not referenced. Which may play a role in the identification of cases.

-Would be beneficial for a little explanation of these datasets used to collect data and how much of the populations are covered by these datasets.

-Figure 1 -4 don't have units on the y-axis. No units in figure 5 either.

-There is notes of ages being higher than those of lower age chort, but I don't see a statistical analysis of these being significantly higher presented. Presented is just a numerically higher rate.

-A comparison of overall mortality rates between countries may be a helpfull comparison chart to create instead of individually by country.

-Conclusion 3 seems to be making conclusions on etiology of rise without the dataset having information to postulate any of these. This should be reserved for discussion not conclusion.

-Conclusion #6 compares cervical cancer to melanoma. There is both vaccine and screening tests for cervical cancer, the comparison to other cancers seems unnecessary.

Minor Comments:

-Introduction before reference 2 does not state which area it is referencing in regards to number of new cases.

- Lines 252-255- I don't see a reference for these sunlight references.

-Discussion doesn't mention if diagnosis difference may play a part in the difference in mortality

-Figure 5 and 8, seems unnecessary to make abbreviations when theres enough room to spell out entire country's name.

Authors present an interesting study of melanoma mortality in Europe. However, the entire discussion is spent speculating on possible etiologies for these differences when it is not possible from the results presented. However, they fail to discuss the other similar studies in the field.

Author Response

Dear Reviewer,

Please find the attached answer.

Sincerely,

Authors

Reviewer 2 Report

This manuscript collected melanoma survival data from 1969 to current records, and analyzed trend in each country. The comparison is interesting and the discussion coordinates some possible reasons for country-specific trends. 

Minor corrections:

1. authors stated: "Melanoma is one of the most preventable cancers. " This is not really true as melanoma incidence continues to increase with the wide use of sunscreen. Indeed melanoma prevention has been failing. 

2. methods: what kind of data was downloaded? counts of death? rates? please specify. If it is counts, then where the population data was gathered? The more details in the method, the better, such as website of those database.

3. line 60-61, (Age standardized rate – ASR: 3,4/100 000). In the same year mela- 60 noma contributes to 57 043 deaths worldwide (ASR: 0,56/100 000) [2].  -- 3,4 and 0,56 should be 3.4 and 0.56, such errors are found in many other places as well.

4. it is interesting to note that Finnish women's mortality started to flat out since 1970, please provide a possible explanation, what is unique about their prevention measures and treatment methods.

5. there are apparent sex difference, please provide the reasons/discussion for this. 

Author Response

(The authors gave the same response as above.)

Reviewer 3 Report

This is a fascinating paper evaluating melanoma mortality trends in European countries.

The presented data is clear and easy to understand. 

My suggestion is to explore in the introduction and especially in the discussion section the new therapies to treat metastatic melanoma, their use in the adjuvant setting, and finally what are the perspectives on the impact on mortality for the following decades. 

And I would add a paragraph at the end of the discussion saying how vital is the presented data and that medical societies have to work hard in educational initiatives for this preventable neoplasia.

Author Response

(The authors gave the same response as above.)

Round 2

Reviewer 1 Report

The authors made no substantive changes and seem to be unwilling to participate in the peer-review process.

While I agree many speculations should be made to interpret the results for the potential causes of changing mortality rates, including these potential etiologies as part of the conclusions overstates the results of the present study.

In response to some of the comments provided:

-The readers of Cancers may be clinical oncologists in addition to epidemiologists. Therefore, a brief explanation of the dataset used is appropriate.

-Units are required in addition to markers on the graph. While there are markers (0-12) the scientific units are not supplied.
